# Sensitivity Modal Analysis of Long Reflective Multimode Interferometer for Small Angle Detection and Temperature

Tania Lozano-Hernandez [1], Julian M. Estudillo-Ayala [1,*], Daniel Jauregui-Vazquez [2], Juan M. Sierra-Hernandez [1] and Roberto Rojas-Laguna [1]

[1]  Departamento de Ingeniería Electrónica, División de Ingenierías Campus Irapuato Salamanca, Universidad de Guanajuato, Carretera Salamanca-Valle de Santiago, Salamanca 36885, Mexico; t.lozanohernandez@ugto.mx (T.L.-H.); jm.sierrahernandez@ugto.mx (J.M.S.-H.); rlaguna@ugto.mx (R.R.-L.)

[2]  Centro de Investigación Científica y de Educación Superior de Ensenada (CICESE), División de Física Aplicada-Departamento de Óptica, Carretera Ensenada-Tijuana, No. 3918, Zona Playitas, Ensenada 22860, Mexico; jaureguid@ugto.mx

*  Correspondence: julian@ugto.mx

**Abstract:** This work presents the sensitive modal analysis of a long reflective multimode optical fiber device for angle and temperature. The reflective multimode interference optical fiber device was fabricated by splicing ~40 cm of multimode optical fiber (50/125). This structure provides a random interference reflection spectrum; the wavelength sensitivity analysis indicates that estimating the angle detection is impossible due to the several modes involved. However, by the phase analysis of the Fourier components, it was possible to detect slight angle deflection. Here, three spectral Fourier components were analyzed, and the maximal sensitivity achieved was 1.52 rad/°; the maximal angle variation of the multimode fiber was 3.4°. In addition, the thermal analysis indicates minimal temperature affectation (0.0065 rad/°C). Moreover, it was demonstrated that there is a strong dependence between the sensitivity and the m-order of the modes involved. Considering the fiber optic sensor dimensions and signal analysis, this device is attractive for numerous applications where slight angle detection is needed.

**Keywords:** angle detection; interferometer; multimode; optical fiber

## 1. Introduction

The study of multimode interference devices has produced several fiber optic sensors [1–5]. This study aims to pursue reliable biosensing, structural health monitoring, and robotics applications [6,7]. For biosensing applications, multimode interference devices offer competitive sensitivity, considering metamaterials [8]. Some of these applications involve slight angle deflection monitoring [9–11]. However, most fiber optic angle sensors are based on interferometric techniques: Mach–Zehnder Interferometer [12,13], Michelson Interferometer [14,15], Fabry–Perot Interferometer [16,17], Fiber Bragg Gratings [18,19], and Tapered Fibers [20–22]. In some cases, their fabrication process is intricate, and the interferometer dimension limits practical applications.

In contrast, multimode interference devices offer no intricate fabrication process and high reproducibility. These devices have been reported in reflection mode operation [4,23,24]. Usually, this optical fiber device uses a short section of the multimode fiber (less than 10 cm). As the length of the multimode optical fiber increases, the losses and the number of modes excited make this structure challenging for some applications; Here, the high sensitivity of long multimode interference devices has been demonstrated for vital signs monitoring [25]. The high sensitivity implies measurement limited and unambiguous measurement range. This trade-off can be overcome by phase analysis [26,27]. Furthermore, this technique allows analyzing the modes involved; here, high-order models are expected to show high sensitivity. Then, this characteristic is used for the sensitive modal analysis of

a long reflective multimode optical fiber interferometer for angle detection. The several modes involved generated a random interference reflection spectrum; as a consequence, the wavelength sensitivity analysis indicates a nonlinear response for angle detection. However, by using the phase analysis, it was possible to detect slight angle deflection linearly. Furthermore, a strong dependence was demonstrated between the sensitivity and the m-order of the modes involved.

## 2. Principle of Operation

As is well-known, when the light from an SMF is coupled to an MMF by a splice point, multiple modes are excited in the MMF section. These modes have different propagation constants; moreover, considering the circular cross-section geometry and the step refractive index profile, the modes generated are linear polarization modes [28,29]. For a reflection mode operation, these modes travel through the MMF's length and are back-reflected by the Fresnel's relation. Then, it is possible to achieve a reflective multimode interferometer (RMMI) when the reflected signal is coupled back to the SMF at the splice point. The reflective multimode interferometer schematic and principle of operation are shown in Figure 1.

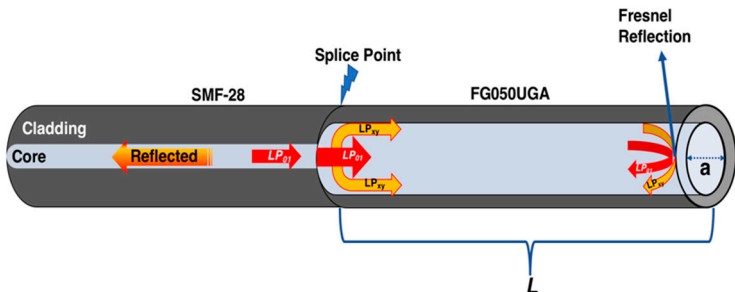

**Figure 1.** Structure diagram of a reflective multimode interferometer.

The excitation coefficient ($C_m$) of the generated modes (*LPxy*) is described by [4,28]:

$$C_m = \frac{\int_0^\infty E_i(r)E_m(r)rdr}{\left(\int_0^\infty |E_i(r)|^2 rdr \int_0^\infty |E_m(r)|^2 rdr\right)^{1/2}} \quad (1)$$

where *Ei*(*r*) and *Em*(*r*) represent the field profile of the incident mode from the SMF and the m-th mode excited at the MMF, respectively; the modes with high coupling coefficients $\eta_m = (C_m)^2$ will generate constructive or destructive interference when they are coupled back at the SMF-MMF interface. This interference depends on the phase between the modes involved ($\Delta\varnothing_{xy}$), and the following relation governs it [4,28]:

$$\Delta\varnothing_{xy} = (\beta_y - \beta_x) * 2L = \frac{\lambda\left(U_y^2 - U_x^2\right)}{4\pi a^2 n_{co}} 2L \quad (2)$$

where $\beta_x$ and $\beta_y$ are the longitudinal propagation constant related to the strongly excited modes ($LP_{0x}$ and $LP_{0y}$) propagated through the double-length (*L*) of the MMF, with a radius "a" and refractive core index $n_{co}$ (see Figure 1). Furthermore, $U_x$ and $U_y$ are the solutions of the zero-order Bessel function. Considering the multiple modes excited, the reflected interference pattern will be irregular. In addition to the several modes involved, the Fourier phase analysis offers the possibility to study the modal sensitivity for each mode and avoid undesired effects from the wavelength spectrum, such as initial set point calibration and cross-talk measurement [30]. Recently, a compressive theoretical and numerical analysis indicated that the strongest radially symmetric modes are mainly involved in the multimode interference effect [31].

## 3. Experimental Setup

An RMMI was fabricated to analyze the modal sensitivity for temperature and angle deflection using the setup shown in Figure 2. This interferometer was achieved using a Fujikura splicer FSM-17S, configured in a multimode program; By this program, the optical fibers are core alignment, and the splicer program parameters are as follows: cleave limit $5.0°$, loss limit 0.20 dB, cleaning arc 150 ms, re-arc time 800 ms, and arc power 20 bit. Here, ~40 cm of FG050UGA fiber is spliced at the end of an SMF-28 section. The interferometer was evaluated for angle detection afterward; the thermal analysis was conducted using chamber Felisa FE-340; here, the RMMI was in a vertical position.

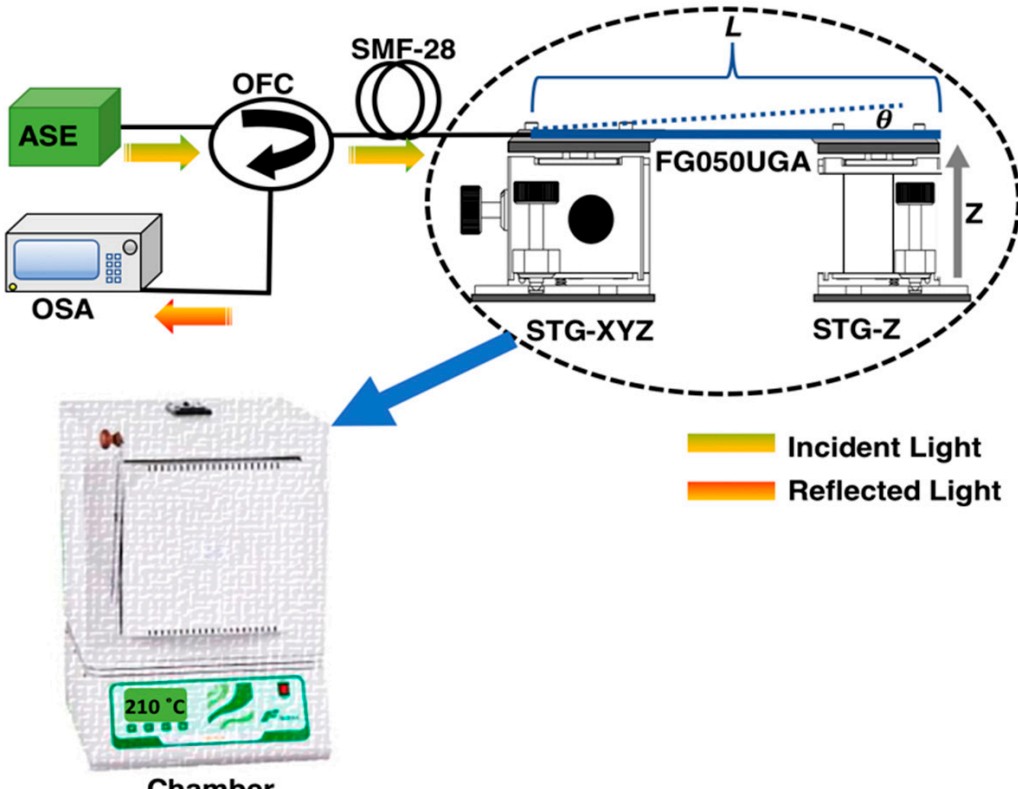

**Figure 2.** Sensing setup for small angle detection based on a reflective multimode interferometer.

The light signal from an amplified spontaneous emission (ASE) broadband source is launched to the reflective multimode interferometer using an optical fiber circulator (OFC). The optical spectrum analyzer (OSA), Anritsu, MS9740A, monitors the reflection signal from the RMMI. The reflective multimode interferometer is set between two translation stages: STG-XYZ and STG-Z. The micro displacements (Z) provided by translation stage STG-Z generate a slight angle to the MMF.

The interference reflection spectrum of the RMMI can be observed in Figure 3. As it was expected, the multiple modes involved generated a random interference reflection spectrum with an oscillating free-spectral range and visibility. By using the Fast Fourier Transform (FFT), it is possible to see the modes involved (please see Figure 3). The Fourier component centered at $0 \text{ nm}^{-1}$ represents the fundamental mode ($LP_{01}$), and the other five spectral components are members of the mode family $LP_{xy}$ [18]. Furthermore, other $LP_{xy}$ members with lower-intensity contributions appear in the spectrum, and all the Fourier peaks contribute to the final interference spectrum shown inset Figure 3.

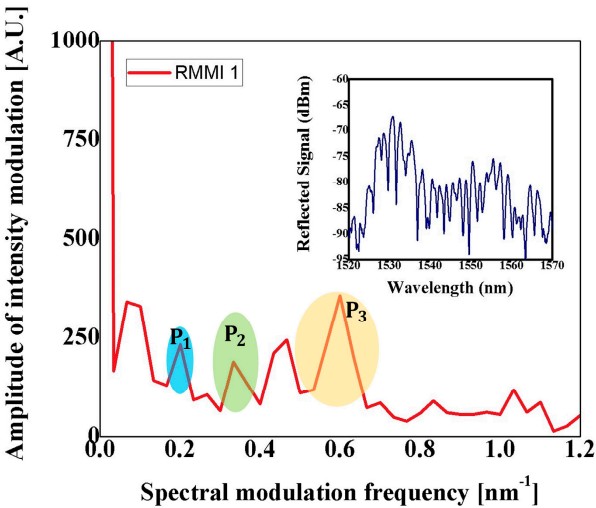

**Figure 3.** Reflection spectrum (inset) and its Fourier transform of a reflective multimode interferometer.

## 4. Results

Figure 4 shows the wavelength reflection spectrum and its Fourier transformation when a slight angle deflection is applied. When the RMMI tilts an angle by using STG-Z, the bending moment (Fixed-beam with sinking support) can be expressed by [32]:

$$M = \frac{6IEZ}{L^2} \qquad (3)$$

where Young's modulus (*E*), the moment of inertia (*I*), the vertical displacement (*Z*), and the MMF fibers' length (*L*) provide information about the strain under the assumption that the stress is proportional to strain, for slight bending and neglecting the shear stresses. Furthermore, due to the elasto-optic effect, the MMF's core suffers a refractive index change by [33,34]:

$$\Delta n_{co} = \frac{n_{co}^3(\varrho_{12} - \nu(\varrho_{11} + \varrho_{12}))\varepsilon}{2} \qquad (4)$$

here, the strain ($\varepsilon$), the Poisson ratio of the fiber ($\nu$) and Pockel's coefficients ($\varrho_{12}$ and $\varrho_{11}$), govern the modal properties described above for the propagated-reflected modes. As a result, the frequency components shown in Figure 4 will be sensitive to the angle deflection as the strain is induced.

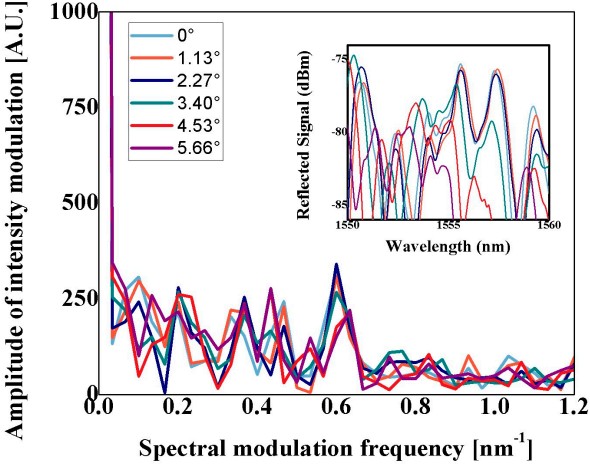

**Figure 4.** Fourier transformation and reflection spectrum (inset) as the angle is applied.

As can be appreciated inset Figure 4, the interference spectrum of RMMI shows a nonlinear response in terms of intensity and wavelength shifting when the deflection angle is increased; Here, for small angles, minimal wavelength shifting can be appreciated. Moreover, when the angle increases, an overlap wavelength spectrum occurs between the initial angles and the maximal angle variation (see inset Figure 4). In addition, the nonlinear wavelength shifting is related to the several modes exited and their interaction among them. In addition, the modes are significantly altered as the slight angle is modified. Furthermore, new modal components are generated, and these components interact with the initial modes. In addition, in Figure 3, it can be appreciated that the distance between the modes is minimal; as a result, they easily interact among them; however, this interaction compromises a linear response in the wavelength spectrum when a physical parameter is applied. It is important to notice that the Fourier spectrum is similar to the expected transmission mode for multimode interferences devices; however, in the reflection modes, the forward-backward light propagation provides a double optical path.

In order to analyze the phase difference contribution of the modes involved, the peak components with minimal changes in their position will be considered for a phase signal analysis. Here, peaks centered at 0.2 (P1), 0.38 (P2), and 0.6 (P3) nm$^{-1}$ of the spatial frequency spectrum will be used for the phase signal analysis (see Figure 3). The phase difference as the angle increase of the mentioned peaks is shown in Figure 5. Three consecutive rounds were performed to evaluate the phase response; here, the angle was increased from 0° to 6°; the phase difference as the angle increased was computed using the technique described by [26,30]; it is essential to notice that the analyzed peaks show a similar polynomial response.

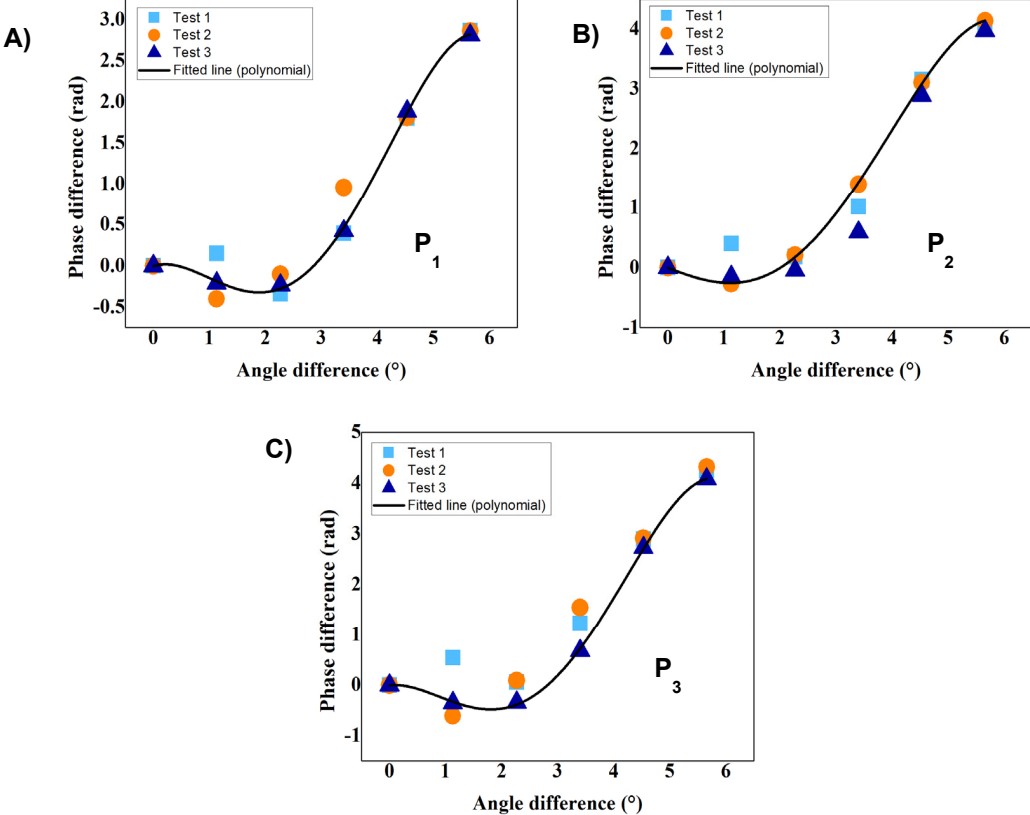

**Figure 5.** Phase difference analysis of spectral frequency components (**A**) P1, (**B**) P2 and (**C**) P3. as the angle is applied at RMMI.

Considering the polynomial response and the positive-negative deviation, the spectral component P1 shows a sensitivity of 0.75 rad/° in the range of 2° to 6° (see Figure 5A). Meanwhile, in the same angle range, the spectral peaks components P2 and P3 show a sensitivity of 1.19 rad/° and 1.52 rad/°, respectively (see Figure 5B,C). As can be observed, the sensitivity increases for high-order modes; these modes are close to the core-cladding interface of the MMF, where significant sensitivity to radial stress can be expected [35].

The thermal analysis is conducted to evaluate the temperature modal sensitivity and the cross-sensitivities for each spectral peak component, showed in Figure 3. The thermal effect over the wavelength spectrum is shown inset Figure 6; here, an apparent wavelength shifting can be observed as the temperature increases for the central wavelength at 1537 nm, then a sensitivity of 1.6 pm/°C can be expected. The linear response can be attributed to the thermo-optic effect and temperature distribution over the long MMF fiber section. It is crucial to notice that the intensity of peak components for the thermal analyses (see Figure 6) shows a slight difference in terms of intensity from the Fourier spectrum shown in Figure 3. Here, it is essential to recall that the RMMI is set vertically under the chamber, then the modes are affected by the process of fixing the RMMI. However, their position remains centered at the same spectral frequency points. Furthermore, after three consecutive rounds, the spectral peak components remain in the same spectral frequency: here, the temperature ranges from 60 °C to 210 °C.

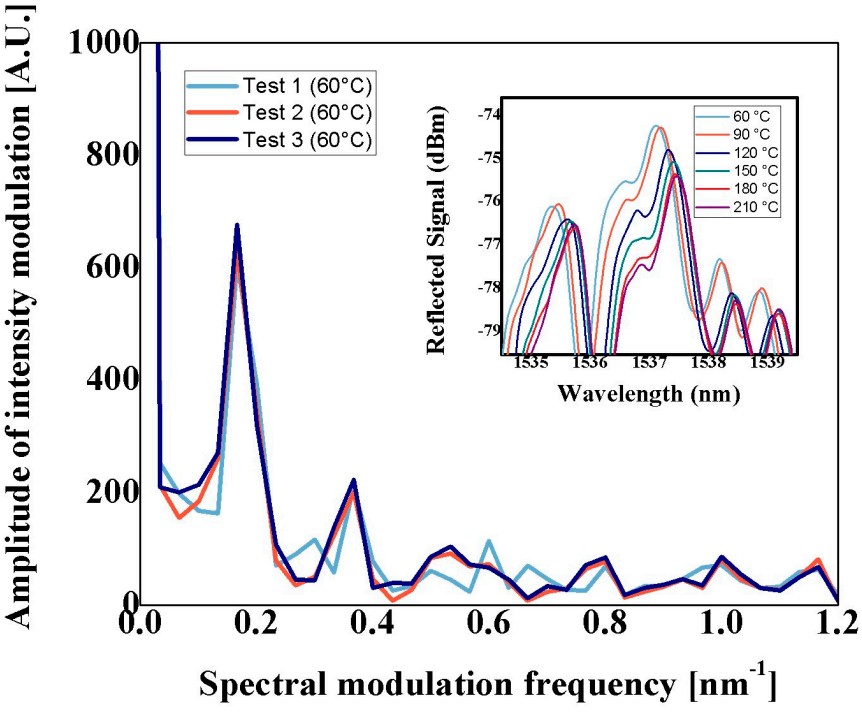

**Figure 6.** Fourier transformation of the RMMI after three consecutive temperature variations at the initial set point and (inset) wavelength reflection spectrum as the temperature increases.

The phase thermal responses of the peaks P1, P2, and P3 are shown in Figure 7; The phase sensitivity of the modal components are 0.003 rad/°C (P1), 0.0062 rad/°C (P2) and 0.0065 rad/°C (P3); As it was demonstrated for angle deflection the high order mode centered at 0.6 nm$^{-1}$ (P3) shows higher sensitivity than other components (P1 and P2). However, this peak component compromises the practical temperature detection; due to its lower intensity, the temperature changes dramatically affect its response (see Figure 7C). The phase response of peaks P1 and P2 are similar (see Figure 7A,B). The cross-sensitivities of the analyzed peaks are 0.004°/°C (P1), 0.005°/°C (P2) and 0.0043°/°C (P3).

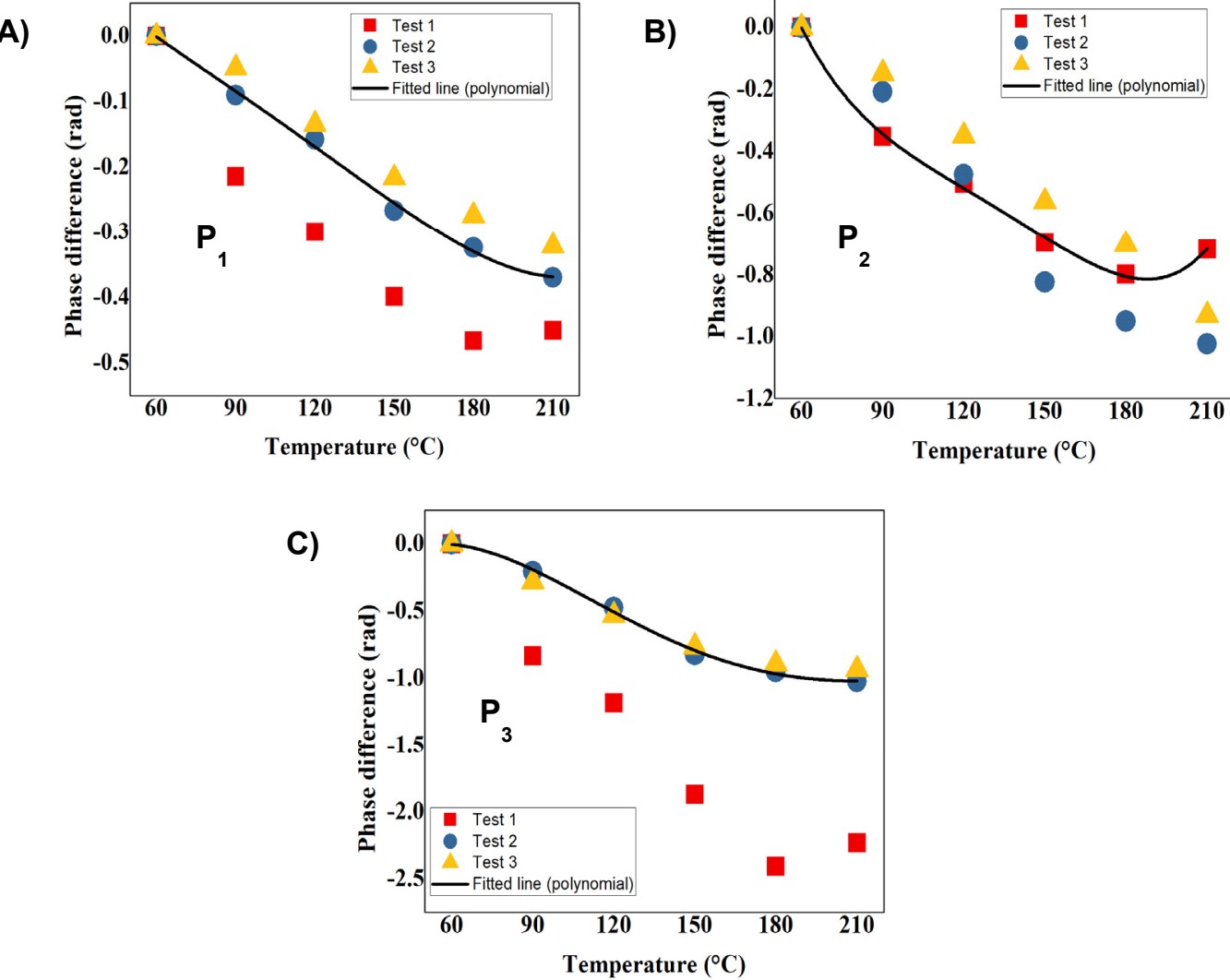

**Figure 7.** Phase thermal response as the temperature increases for (**A**) P1, (**B**) P2 and (**C**) P3.

Considering the high sensitivity for angle detection and the minimal changes observed in the Fourier spectrum during the thermal analysis, a one-way Analysis of Variance (ANOVA) was carried out for P1, P2, and P3 for the angle deflection analysis; here, five consecutive rounds were considered input data. The ANOVA analyses are shown in Figure 8,the above mentioned peaks are analyzed: P1 (Figure 8A), P2 (Figure 8B) and P3 (Figure 8C). As can be appreciated, the mean values for the angle variation between 0 to 2 ° are very similar; however, in the range from 2 to 5°, the mean value increases linearly. Besides, the data are dispersed from the mean value at 4° deflection. A statistically significant difference in the range from 2 to 5° can be expected due to the following *p*-values achieved: $1.013 \times 10^{-14}$ (P1), $3.975 \times 10^{-14}$ (P2), and $2.125 \times 10^{-14}$ (P3), only a few random values were observed for P1 (see red dots inside Figure 8A), and the mean square errors (MSE) were as follows: 0.09 (P1), 0.19 (P2), and 0.30 (P3); as can be observed, their proximity to zero ensures a good prediction for slight angle deflection. The Limit of Detection was estimated by the standard deviation and the sensitivity of each peak; as a consequence, the following values were achieved: 4.88° (P1), 4.22° (P2), and 3.33° (P3). Moreover, the resolution of each peak will be as follows: 1.62° (P1), 1.4° (P2), and 1.16° (P3). As a result, P3 is the optimal spectral peak component for sensing purposes, which agrees with the well-known high sensitivity shown in high-order modes.

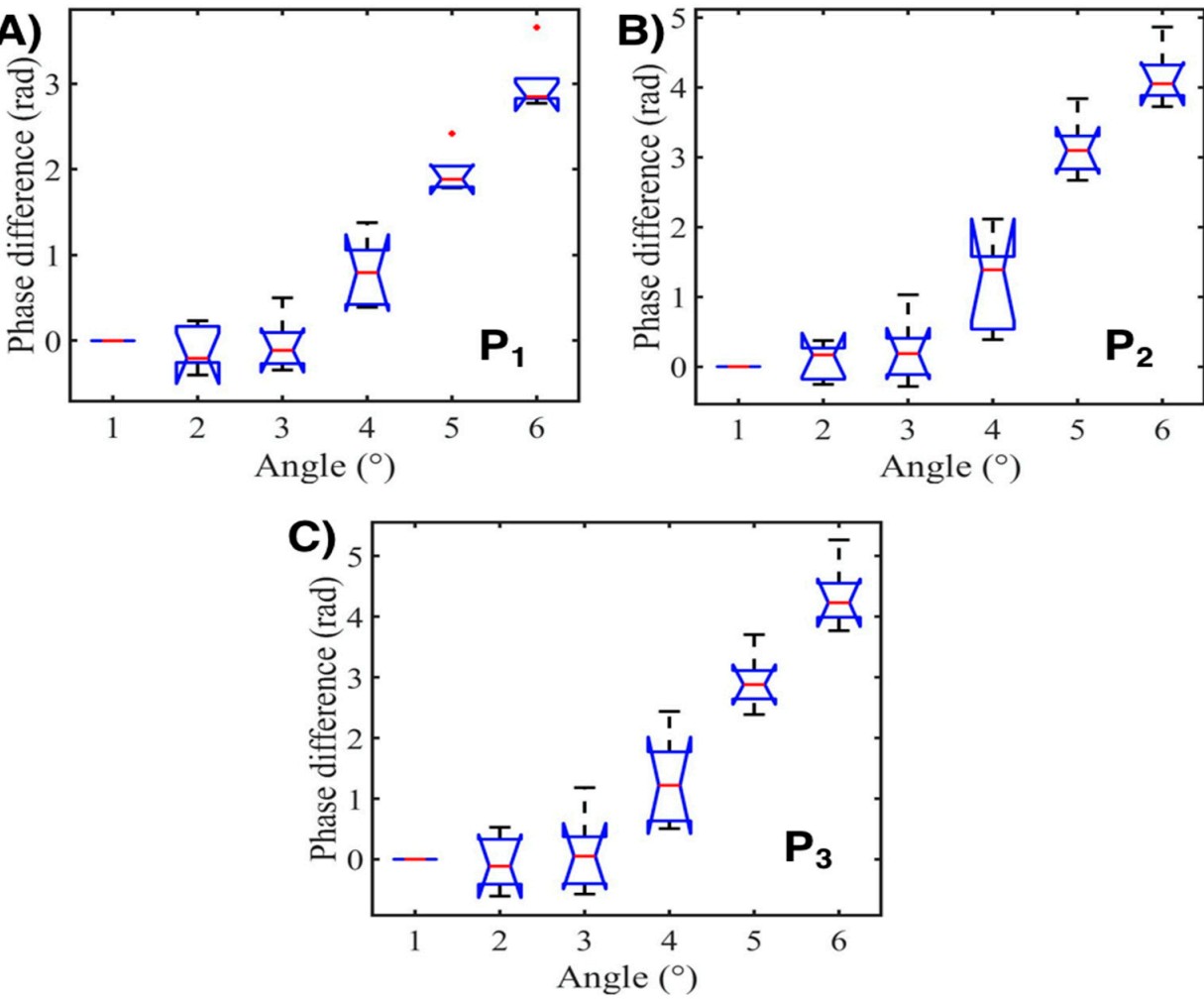

**Figure 8.** One-way ANOVA analyses for (**A**) P1, (**B**) P2, and (**C**) P3 after five consecutive rounds.

With the purpose of validating the stability, the long interferometer was set at the initial deflection angle; afterward, the spectrum was monitored for 30 min using 5 min intervals. This stability analysis is depicted in Figure 9. As can be observed, the peak components suffer minimal changes in their position and during the stability analysis, it can be appreciated the peak components suffer intensity changes; however, their central frequency suffers minimal changes.

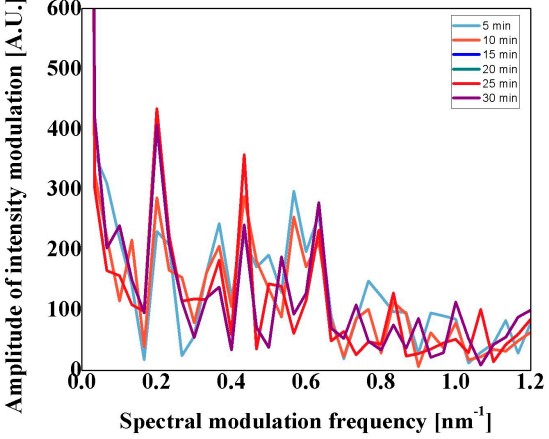

**Figure 9.** Stability analysis of the reflective multimode interferometer.

## 5. Discussion

Considering prior works, the proposed technique and results are competitive; this is summarized in Table 1. The dynamic range of the proposed device can be adjusted by modifying the RMMI's length; as a result, the angle variation can be similar to that reported by [10]. Furthermore, the signal analysis offers high sensitivity, for instance, 1.52 rad/° (P3). The LOD can be improved by using a wide dynamic range; here, the length of the RMMI must be increased. However, this work is devoted to analyses and demonstrating the utility of the Fourier phase analysis for angle and temperature detection.

**Table 1.** Comparative table of some optical fiber structure in terms of sensitivity and physical parameter.

| Optical Fiber Structure | Sensitivity of Angle Detection | Angle Detection Range | Sensitivity of Temperature | Temperature Range | Ref. |
|---|---|---|---|---|---|
| Two Fabry-Perot in parallel | 0.909 nm/° to 35.96 nm/° | $\pm 2.5°$ $\pm 0.2°$ | x | x | [9] |
| Vertical cantilever beam and dual FBGs | $\sim 0.1$ nm/° | $\pm 30°$ | x | x | [10] |
| Tapered fiber Bragg grating | 0.849 dBm/° and 0.583 dBm/° | 0° to 90° | 12 pm/°C | 0 °C to 90 °C | [11] |
| Multimode Interference using Square-Core Fiber | x | x | −15.3 pm/°C | 30 °C to 80 °C | [5] |
| Michelson | 0.55 nm/° | 0° to 50° | x | x | [14] |
| Reflective Multimode Interferometer (RMMI) | 0.75 rad/°, 1.19 rad/° and 1.52 rad/° | 0° to 3.4° | 0.003 rad/°C, 0.0062 rad/°C and 0.0065 rad/°C | 60 °C to 210 °C | This work |

## 6. Conclusions

The modal analysis of a long reflective multimode interference device was conducted for temperature and strain. The reflective device was fabricated by splicing ~40 cm of multimode fiber with a core diameter of 50 μm at the end of the optical fiber circulator. The reflective optical fiber device provides a random interference spectrum with a nonlinear response for angle detection (wavelength spectrum); however, through the phase analysis of the Fourier components, it was possible to detect slight angle deflection. Here, the three spectral Fourier components were analyzed with the following sensitivities: 0.75 rad/° (P1), 1.19 rad/° (P2), and 1.52 rad/° (P3); these sensitivities were estimated for a total angle variation of 3.4°. In addition, the thermal analysis indicates minimal temperature affectation; here, the reflective multimode fiber optic sensor shows the following sensitivities 0.003 rad/°C (P1), 0.0062 rad/°C (P2), and 0.0065 rad/°C (P3), for a temperature range variation of 150 °C. In addition, the thermal analysis indicates that minimal cross-sensitivity can be expected: 0.004°/°C (P1), 0.005°/°C (P2), and 0.0043°/°C (P3). A signal filtering stage will be studied in further work to improve the signal processing analysis. Furthermore, it was demonstrated that high-order modes are more sensitive to the parameters applied. The dimensions and the signal analysis made this device a reliable alternative for small-angle deflection detection.

**Author Contributions:** Conceptualization, D.J.-V. and J.M.E.-A.; methodology, T.L.-H. and D.J.-V.; investigation, J.M.S.-H. and R.R-L.; writing—original draft preparation, D.J.-V. and T.L.-H.; writing— review and editing, D.J.-V. and J.M.E.-A.; visualization, J.M.S.-H.; funding acquisition, J.M.E.-A. and R.R.-L. All authors have read and agreed to the published version of the manuscript.

**Funding:** This research was funded by CONAHCYT Grant A1-S-33363/CB2018.

**Institutional Review Board Statement:** Not applicable.

**Informed Consent Statement:** Not applicable.

**Data Availability Statement:** Under suitable request.

**Acknowledgments:** T. Lozano-Hernandez is grateful to CONAHCYT for the support under scholarship 777601/825870, University of Guanajuato "POA 2023 del Departamento de Ingeniería Electrónica" and CIIC-UG 166/2023.

**Conflicts of Interest:** The authors declare no conflict of interest.

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
