# Peer review of "Sensitivity Modal Analysis of Long Reflective Multimode Interferometer for Small Angle Detection and Temperature"

_photonics, doi:10.3390/photonics10070706_

Round 1

Reviewer 1 Report

The authors theoretically and experimentally studied the sensitivity modal analysis of long reflective multimode interferometer optical fiber device. This work is interesting and has potential applications in many applications where slight angle detection and temperature are needed. I would like to recommend this manuscript for publication, after the following concerns are addressed properly.

[1] In the part of abstract, “1.52rad/°” should be “1.52 rad/°”, and “0.0065rad/°C” should be “0.0065 rad/°C”. The space was missed between the number and unit of physical quantity. There are many similar expressions in the text, which should be corrected before publication.

[2] There are very relevant works about subwavelength gratings: Results Phys. 47, 106354 (2023). These papers can be cited to give audience a broader picture of this field.

[3] The authors believe that “the wavelength sensitivity analysis indicates that estimating the angle detection is impossible due to the several modes involved. However, by the phase analysis of the Fourier components, it was possible to detect slight angle deflection.” Please further explain why the angle detection is impossible by the wavelength sensitivity analysis? And how to further improve the larger angle deflection by the phase analysis of the Fourier components?

Reviewer 2 Report

Authors propose a simple and effective sensor that allow the measurement of temperature and small angle variations. The proposed configuration is based on multimode interference, and the methodology employed is based on the use of FFT. The proposed configuration offers the possibility to measure the small angle changes with good sensibility. In addition, the manuscript was written in a clear and easy to read language.

However, the authors must revise certain aspects before publication can be approved. The following comments are provided:

1.       Please include more recent references

2.       The text does not clarify certain elements of the equation (1). For instance, it is unclear from the text what the terms Cm and ET signify.

3.       Please include more details about the splicing program such as time, current, etc.

4.       Figure 3 depicts the FFT resulting from this experiment. In it, the authors detail all interferometric response contributions. In actuality, each peak represents the interaction or interference of two distinct modes. However, the authors do not specify which modes interfere in this instance. I believe it is significant because it allows us to determine which mode contributes the most to the phenomenon under study. In this regard, I recommend supplementing these significant results with a simulation using FIMMWAVE, COMSOL, Lumerical, or a comparable program to identify the obtained modes.

5.       In Figure 3, the authors examine the variations in phase for P1, P2, and P3 when the sensor is subjected to varying angles. In fact, this is repeated three times (I believe to examine the sensor's repeatability). In this regard, I wished to inquire about the selection criteria for the number of investigations. A meaningful sample size for this experiment would consist of approximately 10 repetitions of the test. In addition, I suggest displaying the mean along with the measurement's standard deviation on a graph.

6.       I think that something that could help them to improve their methodology would be to apply filters and then do inverse transform, I think that with this, they could improve the results by avoiding the contribution of the other interactions.

7.       Please include a comparative table with recent reports.

8.       Please analyze another important parameter in this kind of sensors such as Limit of detection, resolution, etc. In addition, you can analyze the repeatability and stability of the proposed sensor.

Round 2

Reviewer 2 Report

The authors have given a satisfactory response to each observation made. Furthermore, I consider that the quality of the work has increased. For this reason, I approve the publication of the current version.